# Summer Diet of Horses (*Equus ferus caballus* Linn.), Guanacos (*Lama guanicoe* Müller), and European Brown Hares (*Lepus europaeus* Pallas) in the High Andean Range of the Coquimbo Region, Chile

**DOI:** 10.3390/ani11051313

**Published:** 2021-05-03

**Authors:** Giorgio Castellaro, Carla Loreto Orellana, Juan Pablo Escanilla

**Affiliations:** Faculty of Agriculture Sciences, University of Chile, Santiago 8820808, Chile; carla.orellanam@gmail.com (C.L.O.); juanescanillacruzat@gmail.com (J.P.E.)

**Keywords:** dietary overlap, equids, grassland ecology, lagomorphs, mountain summer range, wild South American camelids

## Abstract

**Simple Summary:**

For the proper management of natural grasslands, it is important to know the interactions between the different herbivores, both wild and domestic, that use them. In this research, we studied the botanical composition of the diet of horses, guanacos, and European brown hares, in summer grasslands of the high mountain range of the Region of Coquimbo, Chile. We were able to determine the main species that these herbivores consume, as well as the characteristics of their diets in terms of diversity and quantify the potential trophic competition between them. The main grassland’s species play a maintenance and subsistence role for the three species of herbivores studied, for which they establish selective strategies on certain species of plants, in order to improve the quality of their diets.

**Abstract:**

For an adequate management of natural grasslands, the knowledge and understanding of the dietary habits of herbivores and their trophic interactions are fundamental. During two summer seasons, in a mountain range of a sector of the Coquimbo Region, Chile, the botanical composition, diversity, and similarity of the diets of horses, European brown hares, and guanacos were studied, as was the selectivity of the main grassland plant species, using feces microhistology. The contribution of hydromorphic grasses was similar in the diets of guanacos (35.90 ± 7.27%) and horses (32.25 ± 4.50%), differing from that found in hares (16.32 ± 5.32%). Dryland grassland grasses contributed similarly to the diets of horses (13.21 ± 3.22%), guanacos (22.53 ± 5.21%) and hares (18.35 ± 3.81%), as well as graminoids, which averaged 47.79 ± 6.66%, 35.63 ± 10.76% and 38.94 ± 7.88%, in diets of horses, guanacos, and hares, respectively, without significant differences. The contribution of herbaceous dicotyledons was only important in hares (23.76 ± 3.76%), while that of shrubs was low (<3%) and similar among the three herbivores. Dietary diversity was similar among the three herbivore species (73–79%), with a higher degree of dietary overlap between horses and guanacos (55.7%), which was higher than that obtained between hares and guanacos (50%) and between horses and hares (48%), for which there would be a potential trophic competition between them. The most abundant species of dryland and wet grasslands generally fulfill a functional role of subsistence and a nutritional role of maintenance; however, for the three herbivores studied, a different selective behavior was evidenced, according to their physiological differences, with the selection process little affected by the relative abundance of these species in the grasslands. Due to the above, herbivores resort to the selection of certain species that, despite being not very abundant in grasslands, play an important nutritional and functional role, improving the quality of their diets.

## 1. Introduction

The livestock’s transhumance system in the Coquimbo Region, IV Region of Chile, is used by approximately 20% of the small farmers, who make use of high mountain ranges (“summer ranges”) as the main forage resource [1]. These natural grasslands are used between the summer months of December and March, because for the rest of the year, the climatic conditions prevent it. In these ecosystems, the hydromorphic azonal wet grasslands (“vegas”) are the most important from the point of view of livestock nutrition, because they remain green during the summer and present the highest number of palatable plants species of high nutritional value [2,3]. However, zonal or “dryland grasslands” also contribute, but in a secondary way. In these fragile ecosystems, the knowledge and understanding of the dietary habits of herbivores under grazing conditions is essential to achieve proper livestock management, because it provides information about plants with forage value, diet quality, and potential relationships of competition for the forage resource between domestic and wild herbivores [4]. In the natural grasslands of the high mountains of Coquimbo, domestic horse (*Equus ferus caballus* Linn.) and mule grazing is common, which share the forage resource with wild camelids, mainly guanacos (*Lama guanicoe* Müller), and introduced lagomorphs, such as European brown hares (*Lepus europaeus* Pallas) and wild rabbits (*Oryctolagus cuniculus* Linn.); the characteristics of their diets are unknown, as are the possible negative effects that both domestic livestock and exotic lagomorphs could have on these native camelids. In wet grasslands of Argentine Patagonia (“mallines”), there have been several studies that account for the trophic interactions between domestic cattle and wild herbivores, both exotic (wild rabbits) and native (guanacos) [5,6]. Often, this coexistence would be possible, but generally, native camelids are displaced by domestic livestock towards sub-optimal habitats [7]. This competitive effect may be even more marked when it comes to native ungulates that present a smaller trophic niche width, as pointed out by Vilá et al. [8], in the case of huemul (*Hippocamelus bisulcus* Mol.) and its relationship with sheep and cattle. From the point of view of efficient management of natural grasslands, when estimating the livestock carrying capacity, the existence of this potential trophic competition must be taken into account (in our study, that which would exist between horses, guanacos, and European brown hares). However, estimates of livestock equivalences are often based on the relative intake of the different herbivores, a situation that would only be valid when each animal species consumes exactly the same dietary items, which in nature rarely occurs. For this reason, the calculations of these equivalences should consider, in addition to the dry matter intake of the herbivores involved, the differences in diet composition between them [9,10]. Due to this, the objective of this study was to determine the botanical composition of the diets of horses, European brown hares, and guanacos, analyze their diversity and dietary overlap, and establish the selectivity of the main grassland species, in order to contribute with quantitative antecedents that enable an adequate estimate of range conditions and livestock equivalences when calculating the carrying capacity of the high summer mountain ranges.

## 2. Materials and Methods

### 2.1. Characteristics of Study Area

The study was carried out over two summer seasons (2011 and 2012), in summer ranges of the southern cordillera of Cuncumén, IV Region, Chile (31°52′–32°02′ Lat. S.; 70°19′–70°26′ Long. W.; 3050 m.o.s.l.) (Figure 1).

The area is located in the Steppe Province with a very Cold Dry Summer or Mountain Summer Ecoregion [11], which, according to Köeppen [12], corresponds to the BSsk’ climatic type. This is characterized by being dry, semi-arid (steppe), with concentrated precipitation in winter, where the average annual temperature reaches 6.8 °C, while the monthly average of the warmest month (January) is 11.2 °C, and that of the coldest month (July) averages 2.8 °C. Annual precipitation (rain + snow) is 329.3 ± 153.5 mm (years 1992 to 2019), which is concentrated by 91% between April and September, precipitating in a high proportion as snow (91%). In zonal sites (“drylands”), undulating-soft (10.5–17.5%) to undulating-inclined (17.5–34.5%) slopes are predominant, with coarse, sandy, thin (effective depth close to the 20 cm), not hydromorphic, and rapid drainage soils. In these sites, low bush dominated by subshrubs and “cushion” plants such as *Berberis empetrifolia* Lam. and *Laretia acaulis* (Cav.) Gillies & Hook, respectively, are common. In the herbaceous stratum, species such as *Oxalis compacta* Gillies ex Hooker et Arn., *Poa holciformis* J. Presl, *Bromus setifolius* J. Presl and *Hordeum comosum* J. Presl., are important. In higher sectors, sparse herbaceous formations are common, where rosette species, such as *Menonvillea spathulata* (Gillies & Hook.) Rollins, *Nassauvia lagascae* (D. Don) F. Meigen, *Nastanthus spathulatus* (Phil.) Miers and grasses such as *H. comosum* and *Trisetum preslei* (Kunth) E.Desv., are frequent [13]. The azonal sites (“vegas”) occupy the bottom position of ravines, with a flat-gentle (<4.5%) to gently inclined (4.5–10.5%) slope, soils with a fine sandy-loam to fine sandy texture predominate with effective depths of 50 cm or more, which present a permanent hydromorphism. In these sites, a high cover of hydrophytic vegetation develops, where grasses such as *Deschampsia caespitosa* (L.) P. Beauv. and *Deyeuxia vetulina* Nees & Meyen. stand out. In other sites, and at higher altitudes, cushion plants, such as *Oxychloe andina* Phil and *Patosia clandestina* (Phil.) Buchenau, dominate, as do a series of sedge species of the *Carex*, *Eleocharis* and *Juncus* genera [14].

### 2.2. Methods

In an area of approximately 302 ha, and with the support of the Land Occupation Chart [15], prepared on the basis of a GEoeye satellite image (December 2003; scale 1:3000), two types of grasslands were differentiated: one of the azonal, hydromorphic type (wet-grassland) and one zonal non hydromorphic type (dry-grassland). The latter was delimited based on a buffer area of approximately 500 m around the wet-grassland. In each type of grassland, 28 linear transects of 20 m each were randomly arranged (17 in the wet-grassland and 11 in dry-grassland sites), according to the proportion of each type of grassland present in the evaluated area, with an N–S orientation, where the vegetation cover and botanical composition of grassland evaluations were carried out. The “Modified Point Quadrat” method [16,17] was used, evaluating 100 points spaced every 20 cm on each linear transect. The evaluations were carried out during the first week of March 2011 and 2012. At the beginning of the grassland’s growing season (late December) and associated with each linear transect, an exclusion plot of 4 m^2^ of effective area was installed, which was harvested at the end of the growing season (early March); the material obtained was subsequently dehydrated in a forced air oven at 60 °C for 48 h, in order to estimate the accumulation of dry matter (DM) during the summer growing season. Parallel to the evaluation of the grasslands, the terrain was “cross-country” traveled, collecting as many fresh feces samples as possible of horses (*n* = 17 in the 2011 season; *n* = 18 in the 2012 season), guanacos (*n* = 15 in the 2011 season; *n* = 12, in the 2012 season), and hares (*n* = 15 in the 2011 season; *n* = 18 in the 2012 season), which were analyzed using microhistological analysis [18,19]. From a herbarium of the sector (50 plant species), plant epidermis samples were obtained, applying the techniques proposed by Catán et al. [20] and Castellaro et al. [21], which were later characterized following the criteria and nomenclature proposed by Ortega et al. [22]. In the stool samples of each herbivore, dehydrated for 48 h at 60 °C, ground to 1 mm and discolored with commercial sodium hypochlorite, 100 visual fields were evaluated, using an optical microscope (Olympus, model CX21. Ningbo Huasheng Precision Technology Co. Ltd., Ningbo, China), with built-in digital camera, using 100× and 400× magnifications. The field in which at least one plant fragment was identified was considered a valid field. The identification of epidermis fragments was carried out at the genus and species level when possible. As criteria for identifying cell fragments, characteristics of the epidermal structures were used, such as the shape of the cells, the presence and shape of silica and cork cells, and the presence and shape of trichomes and stomata [21,22]. The relative frequencies of the different species identified was expressed as density, using the tables of Fracker and Brischle [18,22], and based on this last value, the botanical composition of the diet was determined. The plant species present in the diet were grouped into five functional groups: Wet-grassland grasses, dry-grassland grasses, graminoids (Cyperaceae and Juncaceae), dicotyledonous herbs, and shrub species, as proposed by Holechek et al. [4] for rangeland plant management.

Using the data on the diet’s botanical composition, its diversity was determined by calculating the Shannon–Wiener index (H). The previous index was expressed as relative diversity or equality (J):(1)$$H=-\overset{n}{\underset{i=1}{\sum }}P_{i}\cdot {\mathrm{Log}}_{2}\;\left[P_{i}\right]$$
J = H/H_max_(2)

In the above equations, P_i_ is the proportion of the species i in the diet and *n* is the total number of species in the diet. H_max_ represents the value that H would have if all the species found in the diet had the same frequency [23].

With the mean values of botanical composition of the diet obtained for each animal species in the two seasons analyzed, the degree of dietary overlap was estimated, through the calculation of the Kulczynski index (TD) [24]:(3)$$\mathrm{TD}=\frac{\sum_{i=1}^{n}2\cdot W_{i}}{\sum_{i=1}^{n}\sum {\left(a\;+\;b\right)}_{i}}\;$$

In Equation (3), W_i_ is the smallest percentage of a certain plant species when its percentages in the diet of two different animals are compared, and (a + b)_i_ is the sum of these percentages.

The degree of association between the dietary composition of each animal species with the botanical composition of the two types of grasslands was evaluated through the calculation of the Ivlev Selectivity Index (Ei), relating the proportion of a species present in the diet (d_i_) with its proportion in the grassland (p_i_) [25,26]:E_i_ = (d_i_ − p_i_)/(d_i_ + p_i_)(4)
Ivlev index values vary between −1 and 1. Negative values are indicators of rejection towards the species, while positive values indicate preference. Values close to zero reveal indifference to the species in question.

In addition to the previous analysis, the Spearman’s rank correlation was calculated between the values of the diet composition and the botanical composition of the grasslands.

### 2.3. Data Analysis

The data of dry matter production and plant cover of each type of grassland were analyzed independently, assuming the year of evaluation as the only source of variation, using a completely random model in this case. Their botanical composition was analyzed by descriptive statistics, determining the averages of specific contact contribution of plant species. The contribution of the main functional groups of plant species present in the diet, as well as the relevant individual plant species and the dietary diversity index, were analyzed by analysis of variance, assuming a completely random model with factorial structure in this case, considering the animal species, the evaluation season, and their respective interactions as the main sources of variation. In all cases, prior to performing the analysis of variance, their assumptions (independence of the observations, normality, and homoscedasticity) were verified. To detect significant differences, an LSD (Least Significant Difference) Fisher’s test at 95% confidence was used [27]. All of the above analyses were performed using Statgraphics Centurion XVI^®^ software (Statgraphics Technologies, Inc., Virginia, VA, USA).

## 3. Results

### 3.1. Dry Matter Production, Vegetation Cover and Grassland Botanical Composition

No significant differences were found which were attributable to the year of evaluation in DM production in the wet grasslands (*p* = 0.8105), which averaged a summer production of 1013.6 ± 829.7 kg/ha. In the plant cover in these same grasslands, a value of 89.22 ± 12.88% was estimated, and no differences were detected which were attributable to the growing season (*p* = 0.1236). In the dryland grasslands, DM production was lower and with greater variability, with an average of 397.0 ± 308.6 kg/ha, with no differences found which were attributable to the year of evaluation (*p* = 0.1566). The plant cover in these grasslands averaged 27.9 ± 22.7%, without presenting statistical differences between evaluation seasons (*p* = 0.2908) (Figure 2).

The botanical composition of hydromorphic grasslands was dominated by graminoids, where the species *Eleocharis pseudoalbibracteata* (25.2%) and *Carex gayana* (20.2%) stood out. Grasses were also important, especially *Deyeuxia chrysostachya*, which contributed an average of 22.3%. In the dryland grasslands, shrub species dominated (42.3%), in which *Berberis empetrifolia* (12.0%), *Adesmia echinus* (9.0%) and *Nassauvia cumingii* (9.1%) stood out. Dry-grassland grasses represented an average contribution of 31.9%, with *Hordeum pubiflorum* (14.2%) and *Festuca panda* (12.9%) standing out. Dicotyledonous herbs ranked third, with an average contribution of 24.4%, with *Phacelia secunda* being the most important species, with an average contribution of 12.0% (Figure 3).

### 3.2. Diet’s Botanical Composition

The variation in the contribution of grasses from the hydromorphic environment in the diet of the three herbivores studied was significantly explained by the animal species (*p* = 0.0464), with the effect of the season not being important (*p* = 0.646), nor was the animal species × season interaction (*p* = 0.8703). The highest contribution of this group was registered in the diets of guanacos (35.90 ± 7.27%) and horses (32.25 ± 4.50%), with statistically similar percentages among themselves. These percentages differed significantly from those found in the European brown hare’s diet, in which an average of 16.32 ± 5.32% was recorded (Figure 4). Within the group of grasses from the hydromorphic environment (wet grasslands), *Deschampsia caespitosa*, *Deyeuxia chrysostachya* and *Phleum alpinum* were important. The contribution of the first species averaged 5.30 ± 1.49% and 3.00 ± 0.92% in the diet of guanacos and horses, respectively, while in European brown hare’s diets, the contribution was lower (2.60 ± 1.09%). However, the differences were not significant (*p* = 0.3323). Nevertheless, during the first evaluation season (summer 2011), the contribution percentages of this species were significantly higher compared to the 2012 season (*p* = 0.0027). The type of herbivore was significant (*p* = 0.0464) in explaining the contribution of *D. chrysostachya*, a species that was important in the diet of horses and guanacos, with statistically similar percentages of 9.36 ± 2.14% and 7.40 ± 3.46%, respectively. The contribution of this grass in the diet of hares was statistically lower (0.78 ± 2.54%), especially when compared with the diet of horses. Regarding *P. alpinum*, the highest contributions were recorded during the 2012 summer season (*p* = 0.0362), without significant differences between herbivores (*p* = 0.185). Guanacos were the herbivores with the highest contribution of this species (6.13 ± 1.98%), followed by European brown hares (4.90 ± 1.45%) and horses (2.21 ± 1.22%).

Differences were found in the behavior of the xeromorphic environment grasses (dryland grasses), where none of the sources of variation analyzed were significant (*p* > 0.28). This group tended to be more relevant in the guanaco’s diet, with an average contribution of 22.53 ± 5.21%, followed by European brown hare (18.35 ± 3.81%) and horse (13.21 ± 3.22%) diets, which were similar to each other (Figure 4). The species of this group that had the most contribution to the diet of the herbivores evaluated were *Festuca panda*, *Stipa chrysophylla*, *Hordeum pubiflorum* and *Bromus tunicatus*. *F. panda* was relevant in the diet of guanacos (8.16 ± 2.03%), a percentage that was significantly higher than that found in the diet of horses (1.71 ± 1.26%), but statistically similar to the mean determined in the diet of European brown hares (3.96 ± 1.49%), which was also statistically similar to the percentage determined in the diet of horses. In the case of *S. chrysophylla*, the percentages were similar in the three herbivore species, with values of 5.31 ± 1.96%, 2.68 ± 1.43% and 2.27 ± 1.21% in guanaco, European brown hare, and horse diets, respectively (*p* = 0.04182). *H. pubiflorum* showed a different behavior, which averaged 2.89 ± 1.26% in the horse diet, with a higher, although not significant, contribution in the guanaco diet (5.46 ± 2.04%), but significantly lower than the contribution determined in the hare’s diets (7.79 ± 1.50%). However, this last percentage did not differ from that determined in the guanaco diet. The contributions of *B. tunicatus* were statistically similar (although not significant) in the diets of the three species of herbivores evaluated (*p* = 0.2289), with a tendency to be higher in the diets of the horses (5.58 ± 1.34%) and guanacos (2.87 ± 2.17%), compared to the percentage determined in the European brown hares’ diet (2.08 ± 1.58%).

As in the case of dryland grassland grasses, regarding the variation in the contribution of graminoids belonging to hydromorphic grassland, none of the sources of variation analyzed were significant (*p* > 0.36). In the diet of horses, this group of plants averaged a contribution of 47.79 ± 6.66%, being higher, although not significantly, than that found in the diet of hares (38.94 ± 7.88%) and guanacos (35.63 ± 10.76%) (Figure 4). The graminoids with the highest contribution were *Eleocharis pseudoalbibracteata*, *Carex* spp. (*C. gayana*; *C. marima*; *C. vallis-pulchrae*) and *Juncus articus*. The first of these species was important in the diet of horses and guanacos, with similar percentages (16.23 ± 2.83% and 14.67 ± 4.68%, respectively), being significantly different (*p* = 0.0133) from those determined in the diet of European brown hares (2.73 ± 3.75%). The species of the *Carex* genus were relevant in the diet of hares (30.16 ± 4.66%), being significantly different (*p* = 0.0375) with respect to the horses (15.68 ± 3.94%) and guanacos (12.34 ± 6.37%), who averaged statistically similar percentages. In the case of *J. articus* and unlike the other graminoids species, there were no significant differences attributed to the animal species (*p* = 0.1298), with a tendency to be present in a higher proportion in the diet of horses (8.89 ± 2.06%) compared to guanacos (3.21 ± 3.33%) and European brown hares (2.77 ± 2.44%).

The variations in the contribution of the dicotyledonous herb groups were only significantly explained by the effects of the animal species (*p* = 0.0014), being not relevant the effect of the season (*p* = 0.4851) or the animal species × season interaction (*p* = 0.5874). This plant group was important in the diet of European brown hares, where they contributed 23.76 ± 3.76%, a percentage that was significantly higher than that found in the diet of horses and guanacos, whose mean values were statistically similar (5.58 ± 3.18% and 2.94 ± 5.13%, respectively) (Figure 4). Within this plant group, no particular species stood out, except in the case of the hare’s diet, where *Montiopsis potentilloides*, a herb from the zonal environment (dryland grassland), was important, with a contribution of 20.66 ± 3.13%, a percentage that was significantly different (*p* = 0.0004) to that found in the horses (4.19 ± 2.65%) and guanacos (1.04 ± 4.28%) diets, who presented statistically similar values.

Regarding the shrub species of the zonal environment (which were the dominant ones in the botanical composition of dryland grassland), no significant differences were found attributable to the animal species (*p* = 0.3693), season (*p* = 0.3499), or the interaction between both factors (*p* = 0.7871). Only was there evidence of a tendency to present higher percentages of shrubs in the guanaco’s diet (3.00 ± 1.32%) (Figure 4), in particular of the genus *Adesmia* (1.98 ± 0.71%) and *Berberis empetrifolia* (1.02 ± 0.48%), and especially during the summer of 2011.

### 3.3. Diet’s Relative Diversity Index

The studied diets of the herbivores consisted of 20 to 23 plant species. The relative diversity index (J), which jointly considered the richness and the relative contribution of the plant species in the diet, was similar in the three herbivore species (*p* = 0.2087), not being affected by the season (*p* = 0.1411) or the interaction between both factors (*p* = 0.7702). However, the guanaco’s diet tended to be more diverse, compared to those of horses and European brown hares, especially during the 2011 season (Table 1).

### 3.4. Diet Overlap

The dietary overlap, calculated with the mean values of all dietary items for both seasons and for each of the three species of herbivores evaluated, is presented in Table 2.

The highest degree of dietary overlap was obtained between horses and guanacos, while that obtained between horses and European brown hares and between the latter and guanacos were lower and of a similar magnitude.

### 3.5. Selectivity Index of the Main Consumed Plant Species of Wet Grassland

The behavior in terms of selectivity of the main plant species consumed from the hydromorphic grassland is presented in Figure 5, together with the correlation between the Ivlev’s index and the percentage of the different plant species present in the grassland, which was only significant in the case of European brown hares. In guanacos, plant species present in the diet, but whose presence in the grassland was undetectable (E_i_ = 1; absolute selection), were *Deyeuxia erythrostachya*, *Deyeuxia* sp., *Festuca nardifolia*, *Nicoraepoa subenervis*, *Trisetum oreophilum*, *Carex vallis-pulchrae*, *Juncus* sp., *Oxychloe andina*, *Calceolaria filicaulis* and *Gentiana prostrata*. A similar behavior occurred with these plant species in the diet of horses and hares, with the exception of *O. andina*, which was not consumed by lagomorphs. In the case of the diet of equines, *Mimulus depressus* was added to this group. In guanacos, those species present in the grassland, but which were not detected in the diet (E_i_ = −1; absolute rejection), were *Carex maritima*, *Phylloscirpus acaulis*, *Azorella trifoliolata*, *Calandrinia affinis*, *Cardamine vulgaris*, *Cerastium arvense*, *Gayophytum micranthum*, *Lobelia oligophylla*, *Ranunculus cimbalaria* and *Trifolium repens*. In the diet of horses, this group was reduced to the species *Azorella trifoliolata*, *Calandrinia affinis*, *Cardamine vulgaris*, *Cerastium arvense*, *Gayophytum micranthum* and *Ranunculus cimbalaria*. However, in the hare’s diet, *Cardamine vulgaris* and *Cerastium arvense* were selected. In the guanaco’s diet, intermediate selectivity values (−1 < E_i_ < 1) presented the species *Deschampsia caespitosa*, *Deyeuxia chrysostachya*, *Festuca werdermannii*, *Phleum alpinum*, *Carex gayana*, *Eleocharis pseudoalbibracteata*, *Juncus arcticus*, *Patosia clandestina*, *Zameioscirpus gaimardioides* and *Werneria pygmaea*. In the horse’s diet, the species *Carex maritima*, *Phylloscirpus acaulis*, *Lobelia oligophylla* and *Trifolium repens* were added to this group, while in the diet of hares, they were selected, in addition to the previously mentioned species, *Cardamine vulgaris*, *Cerastium arvense* and *Lobelia oligophylla*.

### 3.6. Selectivity Index of the Primarily Consumed Plant Species of Dryland Grassland

In this type of grassland, and for the guanaco’s diet, the species that presented absolute selection were *Bromus tunicatus*, *Trisetum preslei* and *Chaetanthera pulvinata*. In the diet of horses, *Arenaria serpens* and *Arjona patagónica* were added to this group, while in the diet of hares and with the exception of *A. serpens*, *Glandularia sulphurea*, *Perezia carthamoides* and *Phacelia cumingii* were included. The dryland grassland species that were rejected by guanacos were *Chaetanthera euphrasioides*, *Jaborosa laciniata*, *Phacelia secunda*, *Nassauvia cumingii* and *Haplopappus scrobiculatus*, while the horses rejected the same species, with the exception of *J. laciniata*. In the case of hares, only *H. scrobiculatus* was absolutely rejected. Regarding the dryland grassland species that presented intermediate degrees of selectivity (−1 < Ei < 1), in the guanaco diet it is worth mentioning *Festuca panda*, *Stipa chrysophylla*, *Hordeum pubiflorum*, *Montiopsis potentilloides*, *Adesmia* sp., *Berberis empetrifolia* and *Chuquiraga oppositifolia*. In the diet of horses, *J. laciniata* could be added to the previous species group, while in the diet of hares, in addition to the aforementioned species, *Chaetanthera euphrasioides*, *Phacelia secunda* and *Nassauvia cumingii* were selected (Figure 6). In this type of grassland, the correlation between Ivlev’s index and the percentage of the different plant species present in the grassland was significant in the cases of horses and hares. 

Regarding the correlation between the botanical composition of the diets and that of the hydromorphic grassland, none of the three herbivores studied showed significant correlations. However, in the case of the dryland grassland, the guanaco and hare diets were significantly associated with the botanical composition of the grassland (Table 3).

## 4. Discussion

### 4.1. Dry Matter Production, Plant Cover and Botanical of the Grasslands

In summer mountain ranges of central Chile, the information published regarding DM production, plant cover and botanical composition values is scarce. Dry matter productions in the order of 2000 kg/ha have been estimated in hydromorphic grasslands (“bofedales”) in Chilean highlands ranges [28], figures that are higher than those reported in this work. The rainfall amount recorded in those environments could explain the differences. In this regard, Le Houérou [29], reports rainfall use efficiency (RUE, kg/ha/mm/year) in arid grassland ecosystems in good condition that vary between 4 and 6 kg/ha/mm/year. Considering the amount of precipitation recorded in the period prior to the growing season of the years in which this study was carried out (287.6 and 337.2 mm/year), in the wet grasslands an annual DM productivity between 1150 and 1349 kg/ha is estimated, similar to that determined in our research. In wet grasslands of the high mountain range of the Metropolitan Region of Chile, Castellaro et al. [30] reported a high contribution of graminoids in the botanical composition of these grasslands, especially of the *Carex*, *Juncus* and *Eleocharis* genera, with plant cover values varying between 74% and 98%, depending on the grassland condition. This is important, because plant cover has been used as an indicator of degradation in hydromorphic grasslands. In this regard, Ormaechea et al. [31] point out that in the case of wet grassland in the province of Santa Cruz, Argentina, a plant cover percentage lower than 90% would be an indicator of severe deterioration, which suggests that the grasslands in which this study was carried out could show a certain degree of degradation.

In desert grassland, Gamound [32] reported RUE values in the order of 1.9 kg/ha/mm/year), estimating an annual DM productivity in the range of 546 to 641 kg/ha, figures higher than those measured in the dryland grasslands in this work (Figure 2b). The differences could be attributed mainly to aspects related to soil and species composition, factors that define different grassland sites. Botanical composition dominated by shrub species of the genera *Adesmia*, *Chuquiraga* and *Berberis* and plant cover between 26% and 59% have been measured in high-altitude zonal grasslands [30], which would be a characteristic of these ecosystems, adding to it the low productivity per unit of land surface [32].

The fact of not finding significant statistical differences in dry matter production and vegetation cover between the two evaluated summer seasons could partly explain that the “year effect” in most of our analyses did not have significant statistical effects on the variables associated with the composition of diets.

### 4.2. Diet’s Botanical Composition

In the guanaco diet, our research confirms that in the mountain summer environment, this camelid consumes mainly grasses (58.4%), especially those from the azonal environment (wet grassland) which contributed almost 36%, while the dryland grasses contributed 22.5%. Dominance of grasses, especially those that are more fibrous and of relatively lower quality, were reported in the diet of this camelid in the forest-grassland-steppe ecotone of southern Patagonia [33,34], as well as in grasslands of northern Patagonia, Argentina [35]. These species contribute in an important way to their nutrition, because they can be used efficiently given the particular characteristics of the digestive physiology of these camelids [36]. Regarding the graminoid species in the guanaco diet, the percentages obtained in this work were of a similar magnitude to those determined by Arias et al. [34] in the summer diet of guanacos in Tierra de Fuego, Argentina, but higher than those reported by Muñoz and Simonetti [37] and Bonino and Pelliza-Sbriller [38]. This indicates that this group of plant species would be a secondary contribution to the nutrition of this camelid compared to the grasses. The dominance of grass species, followed by graminoids in the guanaco diet, is consistent with that determined by Barri et al. [39] in guanaco populations that have been reintroduced in the Quebrada del Condorito National Park, Argentina. Muñoz and Simonetti [37] and Candia and Dalmasso [40] point out the important contribution of woody species in the diet of this ungulate, an aspect that contrasts with what was obtained in the present study, where the woody component was only 3.0%, a figure lower than that found in the diets of llamas (*Lama glama*), a domestic camelid close to the guanaco [41]. This could be attributed to the low supply of these species in the study area, because despite being important in the botanical composition of dryland grasslands, their low contribution in terms of dry matter would determine a low availability. Added to this is the presence of resinous substances and spiny structures that would affect the palatability of these species. Regarding the contribution of herbaceous dicotyledons to the guanaco diet, several authors point out that this is low [33,34,42], therefore this group of species would have less importance in the nutrition of this herbivore.

In our research, the dominance of graminoids and grasses in the wet grassland in the diet of horses is consistent with the trend indicated by several authors [43,44,45]. Fleurance et al. [46] showed that horses spend a high percentage of their feeding time on humid short-height meadows, similar to those evaluated in this work. This is a way of keeping the grassland plants in a vegetative state in order to achieve a better quality of the diet and a faster transit velocity of the digesta in the digestive tract, in order to compensate for the lower digestion capacity of fibrous elements presented by equines compared to ruminants and pseudo-ruminants [47,48]. A high proportion of grasses (91%), low proportion of shrubs (8%), and almost absence of herbaceous dicotyledons (<1%) in the diet of horses is indicated by Smith et al. [49], partially coinciding with the dietary behavior of the horses evaluated in this work, but reaffirming the grazing behavior of this ungulate. The dietary behavior is repeated from one season to another, which suggests a relative stability in these habits.

In the case of European brown hares, a greater balance between the contributions of grasses and graminoids (with regular to low percentages of crude protein and high fiber content) and dicotyledonous herbs is noteworthy, a trend that remained relatively stable over time. The above is mentioned by Reiclin et al. [50], who point out that dicotyledons and grasses typical of the natural environment dominate in the diet of these leporids, which coincides with the figures provided by López-Cortez et al. [51] in high Andean environments of Chile. However, in other studies [52], the hare is mentioned as a consuming herbivore of graminoids and herbaceous dicotyledons, while Puig et al. [53] and Johnson [9] indicate that grasses are the food source most important for this species. Karmiris et al. [43], in hydromorphic grasslands of the European Mediterranean coast, report a percentage of around 15% for the contribution of dicotyledonous herbs in the diet of hares, while in our study these figures were higher (~24%). This suggests a high importance of this group of plant species in the nutrition of herbivores with cecal fermentation, given the characteristics of their digestive system and the way in which nutrients are used [48]. In boreal environments of Alaska, the American hare (*Lepus americanus*) consumes fir needles, bark, and twigs of birch, willow, fir, and alder, as well as dicotyledonous herbs and deciduous tree leaves [54]. The information provided in this work, as well as that indicated by other researchers, allows us to conclude that these lagomorphs have diverse dietary habits, and are efficient in using diets of low nutritional quality, which contributes to the success of their adaptation to different types of ecosystems.

### 4.3. Diet Diversity

The information obtained in our work regarding the guanaco diet’s diversity shows a wide trophic niche, which could be similar or even higher than that determined in other South American natural grasslands. In this regard, Muñoz and Simonetti [37] in Tierra del Fuego, and Baldi et al. [33] in the province of Chubut, report similar numbers of plant species to those found in our study. This number was higher than that reported by Arias et al. [34], who indicated diets comprising 11 to 13 dietary items. These last authors determined an index of absolute diversity of the diet of this camelid that varies between 1.80 and 2.25, depending on the time of year. Considering the number of plant species present in these diets, these figures generate relative diversities in the order of 0.55, values lower than those obtained in our study. Notwithstanding this, and in accordance with the relative diversity values obtained in this work, a relatively high amplitude of the trophic niche is confirmed, a product of generalist consumer habits.

In the summer mountain ranges of Chile, the diversity of the equine diet has not been studied, especially not comparing it with that of other herbivores that simultaneously graze these grasslands. Our data suggest a similarity in the number of plant species consumed by the three herbivores studied. The number of plant species determined in the diet of horses was higher than that found by McInnis and Vavra [55] in feral horse diets, and that reported by Smith et al. [49] and Krysl et al. [56]. The aforementioned authors do not mention relative diversity values but, given the lower number of species detected (<10 plant species), it is likely that in high mountain ranges of the Coquimbo Region of Chile, the dietary diversity of horses was higher, although as reported by several studies, with a predominance of graminoids and grasses, which coincides with what was indicated by Hosten et al. [57].

Regarding the dietary diversity of the European brown hares, our study shows a high number of consumed plant species (~21), which, together with their proportions, resulted in high dietary diversity, highlighting the intake of a dicotyledonous herbaceous species in particular (*Montiopsis potentilloides*), which was highly selected. The data presented by Uresk [58], in the dietary study of the black-tailed hare (*Lepus californicus*), report a lower number of dietary items (13 plant species), but as in our results, the intake of a dicotyledonous grass (*Achillea millefolium*) [58]. This suggests that this herbivore has a high amplitude of trophic niche, but at the same time presents a selective behavior for some plant species, which would fulfill a nutritional role of production and a functional role as diet improvers [25,59]. Selections of plant species with high energy content associated with relative high levels of fat and protein are reported by Schai-Braun et al. [60], which supports our affirmation. This also coincides with the findings of López-Cortez et al. [51], who point to the European brown hare as a species with greater amplitude of trophic niche compared to native rodents of high Andean environments in Chile, where food availability is scarce. In a similar way to what was reported in our research, the aforementioned authors report a high degree of selectivity for the dicotyledonous herb *Cristaria andicola*. The previous behavior probably relates to a foraging strategy that aims to improve protein levels and/or certain polyunsaturated fatty acids [60], which could be deficient in the ingested diet. The greater dietary diversity of the hare is also reported by Reus et al. [61], who compared the diet of this leporid with that of the native rodent *Dolichotis patagonum* (mara), during the dry season in a desert grassland in the San Juan Province, Argentina.

### 4.4. Dietary Overlap

The experimental evidence obtained in our research related to the dietary overlap between horses, guanacos and European brown hares indicates that this could be of medium magnitude. In accordance with the above and taking into account the dietary overlap ranges suggested by Holechek et al. [4], the grazing capacity of the grasslands where the evaluations were carried out, considering the simultaneous grazing of these three herbivores, would be partially additive, because the degree of dietary overlap is in a range between 30% and 70%. The greater dietary overlap obtained between horses and guanacos would indicate a greater potential competition between these ungulates, compared to the effect that European brown hares could have. In this regard, Baldi et al. [33] report a high dietary overlap between sheep and guanacos, also suggesting a high degree of competition and sympatricity between both ungulates, which is consistent with that reported by Reus et al. [62], regarding the dietary overlap of summer diets between donkeys and guanacos.

However, Linares et al. [63] point out that the dietary overlap between guanacos and horses would be moderate to low, but tending to be greater during the time of abundance of forage, which coincides with what was determined by Puig et al. [35] regarding the dietary overlap between guanacos and cattle. The lowest similarity in the diets between European brown hares and domestic ungulates (sheep and cattle) is also reported by Johnson [9], who points out similarity values of 35% and 45%, when it comes to cattle and sheep, respectively. However, Puig et al. [53] report higher degrees of dietary similarity between European brown hares and horses (56%).

The lower degree of dietary overlap between the hare and the ungulates could correspond to a strategy aiming to minimize the competition for food developed by these leporids, which contributes to their adaptive success. Therefore, the potential trophic competition between the European brown hares and the guanacos would be of less magnitude than that exerted by domestic equines. With the above information, it is possible to suggest livestock equivalences between the three species of herbivores studied, considering, in a first approximation, the metabolic weight ratio [64,65] and the degree of dietary overlap between them [9,10,66]. Assuming mean liveweights of 384, 96 and 4.2 kg, for horses, guanacos, and European brown hares, respectively [67,68,69], as well as the dietary overlap determined in our research, these livestock equivalences can be determined and are presented in Table 4.

In Table 4, it can be observed that 5.08 guanacos of 96 kg of liveweight and 61.32 European brown hares of 4.2 kg would consume the same amount of dry matter (DM) as a horse of 384 kg. In turn, the DM intake of 20.93 European brown hares would be equivalent to that of a guanaco. With these livestock equivalences, a more precise estimation of the grassland carrying capacity could be made, and this calculation could be compared with the current herbivore stocking rate, considering the density of these animals in the area where the evaluation is carried out. This will allow planning grassland management decisions in a more efficient and comprehensive way, considering not only DM intake, but also trophic interactions between herbivores. However, to validate the livestock equivalences proposed in this research, in addition to the aspects related to metabolic liveweight and dietary overlap, other variables should be considered, such as the movement area (home range), territoriality, and spatiotemporal overlaps among domestic and wild herbivores [70].

### 4.5. Selectivity of the Main Plant Species Consumed

Regarding the selectivity of the hydromorphic grassland plant species, there is a group of species, mostly grasses, some graminoids, and dicotyledonous herbs, which were not possible to detect in the grassland botanical composition, but which were present in the diets of the three herbivores studied (E_i_ = 1.0). These species, although their contribution in the grassland was probably low, were selected by herbivores and would be fulfilling a nutritional role of production and a functional role as diet improvers [25]. This type of species contains low fiber content and high nutrient content [28], and probably low contents of antinutritional substances in its tissues [71], which makes it very attractive to herbivores (“ice-cream” species).

Some grasses such as *Deschampsia caespitosa*, *Festuca werdermannii* and *Phleum alpinum*, were detected in the diets of the three herbivores studied, observing high and positive E_i_ values (0 < E_i_ < 1.0), especially in the diet of horses and guanacos, where these plants would play nutritional production roles, acting functionally as diet-improver species. The high E_i_ obtained by these grasses is explained by their low contribution to the botanical composition of the grassland (<2%) and their high presence in the diet. These species should be considered as “desirable” and indicators of excellent condition in hydromorphic grasslands and may be the object of recovery actions to increase their participation in grassland botanical compositions. These species have mean protein and metabolizable energy levels and high fiber content [30], therefore are especially important in the diet of ruminants and pseudo-ruminants, as well as in horses, but they would be of secondary importance in lagomorphs [48].

In contrast, several plant species, despite being present in the grassland, were not detected in the diet of herbivores, which indicates absolute rejection (E_i_ = 0). However, the behavior of certain graminoids, such as *Carex marima* and *Phylloscirpus acaulis*, which despite having a role in the botanical composition of the hydromorphic grassland, were rejected by guanacos, but not by horses and European brown hares, suggests greater pressure from the latter herbivores towards them.

The dominant species of the hydromorphic grassland, *Carex gayana* and *Eleocharis pseudoalbibracteata* in the diet of horses, presented negative E_i_ although close to zero, which would be an indication of a DM intake proportional to its abundance in the grassland, fulfilling a nutritional role of maintenance and a functional role of volume in equines [25,72]. The exception to this behavior was *D. chrysostachya*, a species that tended to be rejected. The guanacos had a different behavior, tending to reject the three species just mentioned, while the European brown hares selected only *C. gayana*. The differential behavior in terms of the selectivity of the dominant species of the grassland suggests establishing different criteria regarding the pastoral evaluation of this type of grassland, which would be dependent on the species of herbivore.

In the dry grassland plant species, several of them were not detected in the botanical composition, thus assuming a low contribution, but despite this they were selected by herbivores (E_i_ = 1.0). This was the case of the grasses *Bromus tunicatus* and *Trisetum presley* and the dicotyledonous herb *Chaetanthera pulvinata*, which could be classified as “delicious” species, fulfilling a nutritional role of production and a functional role as diet improvers [25]. The *Arjona patagonica* herb would fulfill a role similar to that of the aforementioned species, but only in the diets of horses and European brown hares, while the herb *Arenaria serpens* would fulfill these roles only in the diet of horses.

Contrary to this behavior, many of the dicotyledonous herbs from the dryland environment were rejected by guanacos and horses, such as *Chaetanthera euphrasioides*, *Phacelia secunda* and the shrub *Nassauvia cumingii*, even though the latter was relatively abundant in this type of grassland. It is probable that these species may present certain antinutritional compounds, which could be detected by herbivores and thus induce rejection behavior [25,71,73].

The grass *Stipa crishophylla* and the dicotiledoneous herb *Montiopsis potentilloides* were selected, despite their low abundance in dry grasslands (<4%). The first plant species for guanacos and horses, while the second, for European brown hares. In the leporids, the intake of *M. potentilloides* could be explained by the fact that these plant species could contain certain essential nutrients in high concentrations, and that when selected, would contribute to a more balanced diet [60]. In the case of equines and guanacos, the selection of *S. crishophylla* could contribute to achieving the minimum effective fiber necessary for the health of the digestive tract and the proper functioning of the fermentative process at the pseudo-rumen and colon level, respectively [48,72].

In the diet of horses, the main grasses of the dryland grassland, *Hordeum pubiflorum* and *Festuca panda*, would fulfill a nutritional maintenance role and a functional volume role, because they were consumed in proportion to their abundance [25,72]. However, in guanacos, *F. panda* was selected, while *H. pubiflorum* was consumed in proportion to its abundance. For their part, European brown hares selected *H. pubiflorum*, while *F. panda* tended to be rejected. Regarding the most abundant shrubs, *Berberis empetrifolia* and *Chuquiraga oppositifolia*, a tendency towards rejection was also observed, being of less magnitude in the guanacos where these shrubs would fulfill a nutritional role of maintenance and contribute volume to the diet, as well as *C. oppositifolia* in the diet of horses. However, in these ungulates, *B. empetrifolia* would only contribute to subsistence and survival, while in European brown hares, both shrubs would fulfill these roles (Figure 6). The different selective behaviors observed in the three most abundant plant species of dryland grassland by the herbivores could be associated with the differences in digestive efficiency [48] and their interaction with attributes of these plant species, especially associated with the content of fiber, protein, and resins.

In wet grasslands, and in the case of horses and guanacos, it is important to highlight the absence of positive correlation between the selectivity of the species and their relative abundance in the grasslands, which would be an indication that the selective process of these herbivores would be regulated mainly by the nutritional quality of the plant species, independent of their relative abundance in the grassland [74]. This behavior was different in dry grasslands, where only in horses such independence existed. In the case of European brown hares, both in wet and dryland grasslands, there was a tendency to show higher selectivity when the relative abundance of plant species in the grassland was lower.

Finally, it is important to indicate that only guanacos and European brown hares showed a significant correlation between the botanical composition of their diets and the botanical composition of the dry grasslands (Table 3), which would be an indicator of a certain preference for the plant species of this type of grassland and less dependence on species from the hydromorphic grassland.

## 5. Conclusions

Our results suggest that in the mountain range environment of the Coquimbo Region of Chile, the guanaco behaves as an essentially bulk and roughage eater, consuming a high amount of grasses. This contrasts with the diet of horses, in which there is a predominance of graminoids from the wet grasslands, while in European brown hares, in addition to graminoids, there is a high intake of dicotyledonous herbs. The three species of herbivores have a high and similar trophic diversity, which shows generalism in their dietary habits, with a significant dietary overlap, which could imply potential competition and partially additive grazing capacity between them, especially among ungulates. The most abundant plant species, belonging to both humid and dry grasslands, generally fulfill nutritional maintenance roles and functionally act by adding volume to diets. The herbivores studied had different selective behaviors, according to their physiological differences, with the selection of the different plant species hardly affected by the relative abundance of plant species in the grassland. As a result, herbivores resort to the selection of certain species that, despite being not very abundant in grasslands, play important nutritional and functional roles, helping to improve the quality of their diets. Only in the cases of guanacos and European brown hares was a certain preference for the dryland environment evidenced, because their diets were positively correlated with the botanical composition of this type of grassland.

## Figures and Tables

**Figure 1 animals-11-01313-f001:**
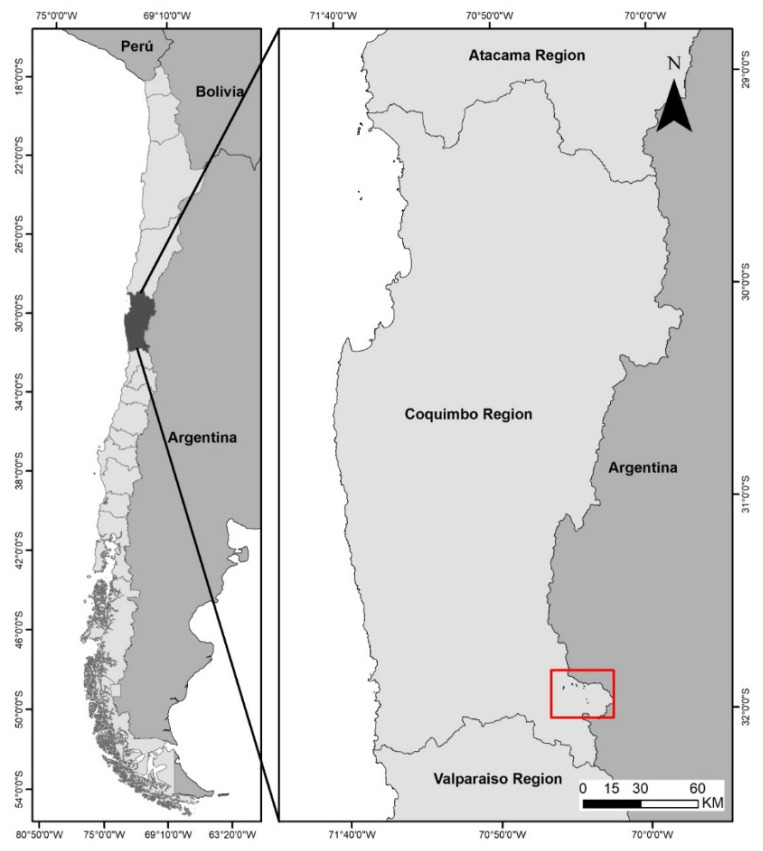
Location of the area where the study was carried out.

**Figure 2 animals-11-01313-f002:**
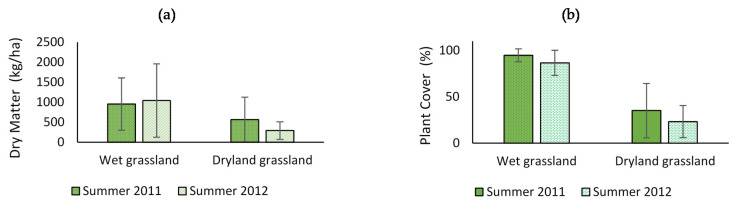
(**a**) Dry matter production and (**b**) plant cover in wet and dryland grasslands in the study area, during the two evaluation seasons. Lines above and below the average indicate one standard deviation.

**Figure 3 animals-11-01313-f003:**
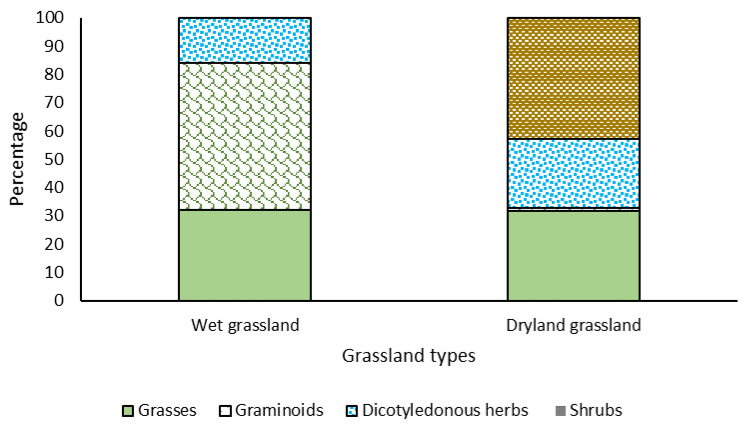
Percentage contribution of the main functional groups of plants to the botanical composition of wet and dryland grasslands present in the study area. Average of two evaluation seasons.

**Figure 4 animals-11-01313-f004:**
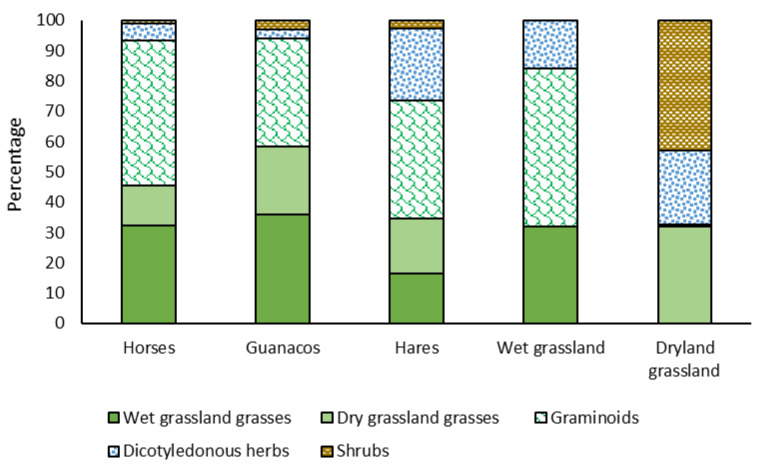
Relative contribution of grasses, graminoids, dicotyledonous herbs, and shrub species, to the diets of horses, guanacos, and European brown hares that graze high mountain ranges of the Coquimbo Region, Chile. The same figure includes the mean contributions of the same functional groups in the botanical composition of the grasslands. Average of two evaluation seasons (2011 and 2012).

**Figure 5 animals-11-01313-f005:**
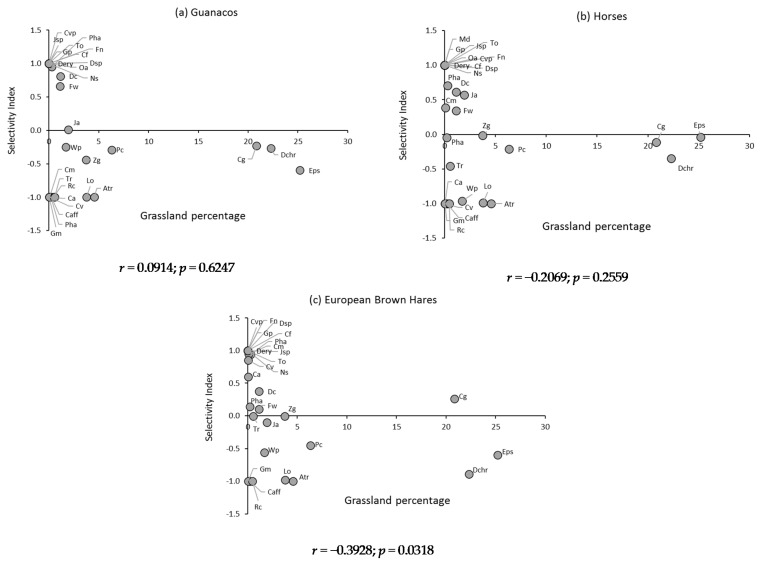
Ivlev’s selectivity index for the main plant species of (**a**) guanacos, (**b**) horses, and (**c**) European brown hare’s diets, in hydromorphic grasslands. Average values of two summer evaluation seasons. *Deschampsia caespitosa (Dc)*, *Deyeuxia chrysostachya (Dchr)*, *D. erythrostachya (Dery)*, *D. (Dsp)*, *Festuca werdermannii (Fw)*, *F. nardifolia (Fn)*, *Nicoraepoa subenervis (Ns)*, *Phleum alpinum (Pha)*, *Trisetum oreophilumv(To)*, *Carex gayana (Cg)*, *C. maritima (Cm)*, *C. vallis-pulchrae (Cvp)*, *Eleocharis pseudoalbibracteata (Eps)*, *Juncus arcticus (Ja)*, *J. (Jsp)*, *Oxychloe andina (Oa)*, *Phylloscirpus acaulis (Pha)*, *Patosia clandestine (Pc)*, *Zameioscirpus gaimardioides (Zg)*, *Azorella trifoliolata (Atr)*, *Calandrinia affinis (Caff)*, *Calceolaria filicaulis luxurians (Cf)*, *Cardamine vulgaris (Cv)*, *Cerastium arvense (Ca)*, *Gayophytum micranthum (Gm)*, *Gentiana prostrata (Gp)*, *Lobelia oligophylla (Lo)*, *Mimulus depressus (Md)*, *Ranunculus cymbalaria (Rc)*, *Trifolium repens (Tr)*, *Werneria pygmaea (Wp).*

**Figure 6 animals-11-01313-f006:**
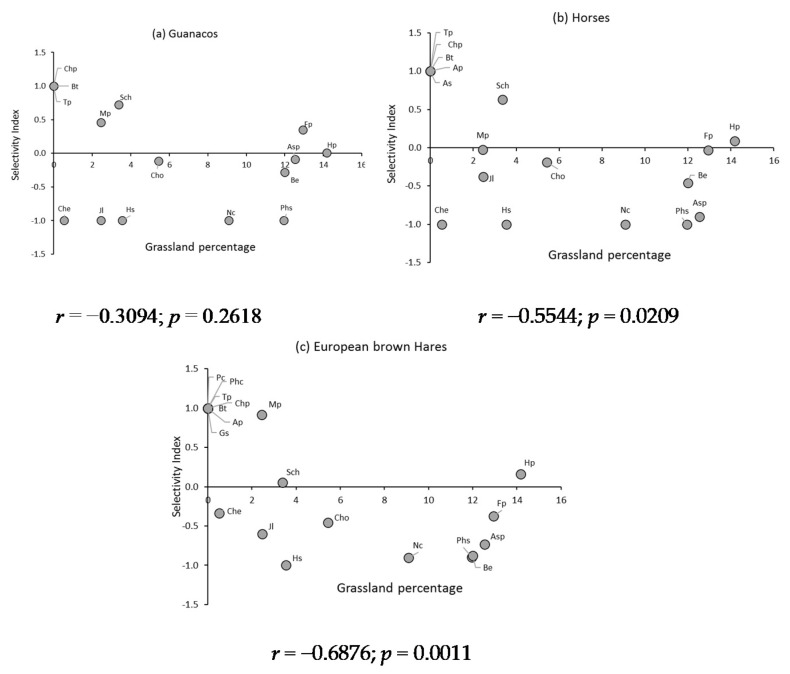
Ivlev’s selectivity index for the main plant species of (**a**) guanacos, (**b**) horses, and (**c**) European brown hare’s diets, in dryland grasslands. Average values of two summer evaluation seasons. *Bromus tunicatus (Bt)*, *Festuca panda (Fp)*, *Stipa chrysophylla (Sch)*, *Trisetum preslei (Tp)*, *Hordeum pubiflorum (Hp)*, *Arenaria serpens (As)*, *Arjona patagonica (Ap)*, *Chaetanthera euphrasioides (Che)*, *Chaetanthera pulvinata (Chp)*, *Glandularia sulphurea (Gs)*, *Jaborosa laciniata (Jl)*, *Montiopsis potentilloides (Mp)*, *Perezia carthamoides (Pc)*, *Phacelia cumingii (Phc)*, *Phacelia secunda (Phs)*, *Adesmia sp. (Asp)*, *Berberis empetrifolia (Be)*, *Chuquiraga oppositifolia (Cho)*, *Nassauvia cumingii (Nc)*, *Haplopappus scrobiculatus (Hs).*

**Table 1 animals-11-01313-t001:** Dietary diversity index (minimum square average ± standard error) of horses, guanacos, and European brown hares grazing on high mountain ranges of the Region of Coquimbo, Chile, for two seasons.

Season	Herbivore Species			Mean
	Horses	Guanacos	European Brown Hares	
2011	0.797 ± 0.028	0.824 ± 0.033	0.737 ± 0.037	0.786 ± 0.019
2012	0.749 ± 0.026	0.759 ± 0.052	0.722 ± 0.026	0.743 ± 0.021
Mean	0.773 ± 0.019	0.791 ± 0.031	0.730 ± 0.022	

**Table 2 animals-11-01313-t002:** Dietary overlap index between horses, guanacos, and European brown hares grazing on high mountain ranges of the Region of Coquimbo, Chile. Averages were obtained over two seasons.

Herbivore Species	Herbivore Species		
	Horses	Guanacos	European Brown Hares
Horses	-	0.5571	0.4822
Guanacos	0.5571	-	0.4995
European brown hares	0.4822	0.4995	-

**Table 3 animals-11-01313-t003:** Spearman’s rank correlation coefficients between the botanical composition of the guanacos, horses, and European brown hares’ diet and the botanical composition of grasslands. For each case, the number of pairs of observations (*n*) and the statistical significance of the correlation (*p*-value) are included.

Herbivore Species	Wet Grassland	Dryland Grassland
	0.0467	0.5341
Guanacos	*n* = 41	*n* = 23
	*p* = 0.7676	*p* = 0.0122
	0.2030	0.3650
Horses	*n* = 41	*n* = 23
	*p* = 0.1991	*p* = 0.0869
	0.0801	0.5734
European brown hares	*n* = 41	*n* = 23
	*p* = 0.6126	*p* = 0.0072

**Table 4 animals-11-01313-t004:** Livestock equivalences between horses, guanacos, and European brown hares grazing high mountain ranges of the Region of Coquimbo, Chile.

Herbivore Species	Livestock Equivalences ^1^		
	Horses	Guanaco	European Brown Hares
Horses (384 kg liveweight)	-	5.08	61.32
Guanaco (96 kg liveweight)	0.20	-	20.93
European brown hares (4.2 kg liveweight)	0.02	0.05	-

^1^ Livestock equivalence (X) calculated with the equation proposed by Vallentine [10]: $X=\frac{{{(W}_{g}{/W}_{p})}^{0.75}}{\mathrm{TD}}$, where W_g_ is the liveweight of the largest herbivore, W_p_ is the liveweight of the smallest herbivore, and TD is the degree of dietary overlap (fraction) existing between the herbivores being compared.

## Data Availability

The data presented in this study are available on request from the corresponding author.

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
