# Peer review of "Summer Diet of Horses (Equus ferus caballus Linn.), Guanacos (Lama guanicoe Müller), and European Brown Hares (Lepus europaeus Pallas) in the High Andean Range of the Coquimbo Region, Chile"

_animals, 2021, doi:10.3390/ani11051313_

Round 1
Reviewer 1 Report
Review
It should be said that the authors undertook quite interesting research that should be of application and practical importance. In the current environmental conditions, where we are dealing with an evident and progressive intensification of agricultural production, the issues of using the natural environment by wild animals, and at the same time grazing in the same areas of farm animals, are becoming quite important. This is quite important both for trophic and behavioral reasons related to habitat preferences and territorialism.
However, after reading the manuscript, I had some remarks:
The first one is the question of comparing the diets of three species, preferring similar areas of trophic specificity, but differing diametrically with the digestive specificity of the consumed food and, perhaps most of all, with the amount of food demand. After all, horses are monogastric animals, llamas are pseudo monogastric animals, because they are subject to the phenomenon of chewing (anatomically and physiological digestive tract adapted to chewing food), hares are monogastric animals with specific digestion manifested by cecotrophy.
In my opinion, this is a completely wrong methodological approach.
Nevertheless, the comparison of the diets of horses and llamas may be justified in a way, because indeed, as the authors themselves state, these species share the food resources of some environments, and the intensification of breeding displaces some wild species into habitats with a much poorer food base.
Detailed comments:Line 84, taking into account the description of this part of the subchapter, should rather have the name "Characteristics of the research area".Line 119, I do not know why, since there were 28 transects - 17 were wetlands and 11 were dry grasslands. What did this result from, is it the proportionality of the sample size?Line 139, which means: at least one plant fragment identified? Whether it is plant food in general or a plant fragment, but assigned to a specific species specificity. This fragment of the methodology should be detailed.Line 144-146, what was the reason for this division?Line 212-220, this is a repetition of the methodological part, but more detailed, it must be connected with the methodology as a whole.Line 229-232, the initial part of the results confirms my earlier assumptions that the selection of species for comparison was not very accurate.Line 242-232, as above.Line 264-267, unfortunate wording in the comparison where the reader may conclude that the hares are also ungulates (!). Moreover, in my opinion, the lack of statistical significance of the share of B. tunicatus in the diet of horses and the other two species, especially hares, where it is over 2.5 times lower compared to the share of horses (!)Line 277-283, quite an interesting result concerning such a high proportion of sedges, i.e. plants with a very low content of proteins and other nutrients, in other environmental conditions practically not tolerated by hares. This is another proof that the comparison of these species is not fully justified.Line 295-301, confirmation of my earlier statement. However, has it been noted that the authors themselves have shown the variation in the share of vegetation cover (Figure 1), and thus also the availability of dry matter in dry habitats in the summer of 2012?Line 388-391, triviality, after all it is logical that in the case of arid habitats the botanical composition of these habitats deteriorates, and thus the composition of the diet for the species found there becomes almost forced. This is of particular importance in species with small home ranges, such as hares. However, the authors downplayed this problem.Line 401-409, in my opinion redundant information about the total production of biomass (dry matter), unsuitable for the results obtained.In fact, the entire subsection 4.1 Dry matter produccion, plant cover and botanical of the grasslans is redundant. Some information can be used in the methodical part of the work to make a more detailed description of the research area.Line 430-432, these are the results, so why are they being repeated in the discussion?Line 457-458, again referring to the detailed results, it was already in the results, there is no need to repeat.Line 470-471, a result reference again, this is a discussion, not a sequential reporting of results.Line 490, as above.Line 501, as above.Line 536-537, confirmation of my earlier comments about the inappropriate selection of species for comparison.Line 560-562, (Table 3), I have considerable doubts as to this comparison of the dietary equivalent, if only because of the dietary diversity shown in the article, as well as the ranges of occurrence (territorialism) in individual species that are radically different. In my opinion, such a comparison in this case is not legitimate.Line 574-576, another question whether we can talk about trophic interactions between hares and two other species, despite the fact that they live in the same area. In my opinion no. This thesis is also confirmed by the fact articulated by the authors, in the following part, that the diet contained species that were not detected in the botanical composition of meadows, but were nevertheless detected in the food. There were also species found in botanical composition rather than in the diet. This proves the outstanding selectivity of these few species, as well as the possibility of their selection only by certain species of animals.Line 669-671, triviality, perhaps logical that a herbivorous animal feeds on plants (!).The conclusion is only a sumptuous summary of the results. In my opinion, it lacks further guidelines on the management of wild animal populations and grazing of farm animals in the same areas (diet overlap), and yet that was one of the goals of this article (as I guess). Generally, after making the corrections and additions described in the comments and a partial rebuilding of the article, I recommend the manuscript for printing.
Author Response
Dear Reviewer:
You will be able to find the answers to your observations by consulting the attached file.
Thank you so much

Reviewer 2 Report
- Line 60. “Hares” should be “European brown hares”, “rabbits” should be “wild rabbits”.
- Lines 77-82. The end of the introduction lacks clear aims and predictions.
- Study area. Please, add a map so to help who is not familiar with South America to locate your study site.
- Lines 140-147. Please, clarify how you identified plant species, provide some more details on how you prepared slides and how you compared with references.
- Figures 1-3; colours should be different so to allow colour-blind readers to appreciate differences.
- Figure 4 is not auto-explicative. I suggest authors to add the meaning of codes also in the figure caption. Please, divide the figure into 4a, 4b and 4c to for clarity.
- Discussion should be reorganized, starting with showing the novelty of your work, your results and what you found. Then, move to comparison with previous studies.
- As to competition between ungulates and hares, I suggest you to read and cite this paper Viviano A., Mori E., Fattorini E., Mazza G., Lazzeri L., Panichi A., Strianese L., Mohamed W.F. (2021). Spatiotemporal overlap between the European brown hare and its potential predators and competitors. Animals 11: 562.
Author Response

(The authors gave the same response as above.)

Round 2
Reviewer 1 Report
I am glad that in many cases the authors agreed with my comments and suggestions, which certainly allowed the manuscript to be improved. However, after rereading the manuscript, I came up with a few remarks that, in my opinion, need to be taken into account, which will further improve the quality of the manuscript. :
Here they are
Line 15, there is no need to use the full species name in the abstract, it should be stated in the title and once in the abstract.
Line 25, as above
In the following chapters of this work, in my opinion, there is also no need to use the full species name of hares and other species, if it has already been articulated once before. It is unnecessary to expand the text of the work. After all, it is known that it is still the same genre.
Line 228 236, I still maintain my position from the first review that this is a refinement of the methodological part, it needs to be changed.
Line 295-301, I still have doubts about these sedges. Indeed, it is difficult to disagree with the authors' explanations that the differences in the digestive tract that affect the physiological course of digestive processes affect the feeding preferences of these animals. In their explanations, the authors state that the average crude protein content of these grasslands is 8.4%. However, they still have not commented on the sedges. In my opinion, this may be a false result of such a high proportion of these plants in the diet. This must be absolutely verified.
Line 313-317, I still have doubts. The authors did not convince me by their explanations, referring to the statistical results, where statistical differences between the seasons were found on dry grasslands? This needs to be verified.
Line 580-592, I still do not agree with the authors' statements that the purpose of their research was to determine the similarity of eating habits, because they cite the literature on the comparison of ungulates diet, which confirms the food competition of the two analyzed species, but in my opinion the hare (regardless from population density indicators) can in no case be a food competitor for ungulates. The concepts of food habits and food competition are not the same. This must absolutely be changed.
Generally, after making the next few minor corrections and additions described in the comments, I recommend the manuscript for printing.
Author Response
Dear Reviewer
Thank you very much again for reviewing our manuscript.
the response to your comments, you can see them in the attached file
Sincerely,

Reviewer 2 Report
The paper is really improved and can be accepted for publication.
Author Response
Dear Reviewer
Thank you very much again for reviewing our manuscript.
Sincerely,